# Bi-directional Goal-Conditioning on Single Policy Function for State Space Search

## Abstract

State space search problems have a binary (found/not found) reward system. However, in the real world, these problems often have a vast number of states compared to only a limited number of goal states. This makes the rewards very sparse for the search task. On the other hand, Goal-Conditioned Reinforcement Learning (GCRL) can be used to train an agent to solve multiple related tasks. In our work, we assume the ability to sample goal states and use the same to define a forward task ($\tau^*$) and a reverse task ($\tau^{inv}$) derived from the original state space search task to ensure more useful and learnable samples. We adopt the Universal Value Function Approximator (UVFA) setting with a GCRL agent to learn from these samples. We incorporate hindsight relabelling with goal-conditioning in the forward task to reach goals sampled from $\tau^*$, and similarly define 'Foresight' for the backward task. We also use the agent's ability (from the policy function) to evaluate the reachability of intermediate states and use these states as goals for new sub-tasks. Further, to tackle the problem of reverse transitions from the backward trajectories, we spawn new instances of the agent from states in these trajectories to collect forward transitions which are then used to train for the main task $\tau^*$. We consolidate these tasks and sample generation strategies into a three-part system called Scrambler-Resolver-Explorer (SRE). We also propose the 'SRE-DQN' agent that combines our exploration module with the popular DQN algorithm. Finally, we demonstrate the advantages of bi-directional goal-conditioning and knowledge of the goal state by evaluating our framework on classical goal-reaching tasks, and comparing with existing concepts extended to our bi-directional setting. Our implementation can be found here.

## 1 Introduction

The combination of Reinforcement Learning Sutton & Barto (2018) with the strength of highly expressive function approximation offered by Deep Learning (Goodfellow et al., 2016) has achieved multiple breakthroughs in a wide range of control tasks such as in-hand robotic manipulation [(Andrychowicz et al., 2018),(Nagabandi et al., 2019)], semantic object picking (Kalashnikov et al., 2021) and walk gait generation (Haarnoja et al., 2018b) as well as long-range sequential decision-making tasks such as the classic game of Go (Silver et al., 2016), Chess (Silver et al., 2017), Dota 2 (Berner et al., 2019), Atari games (Mnih et al., 2013) and many others. However, these successes have depended on hundreds of millions of samples and data points. Despite Deep Reinforcement Learning having huge potential for solving many real-world practical problems that are modelled as an MDP, the problem of sample efficiency is a big hurdle for Deep-RL researchers.

An RL agent learns how to solve a task based on the reward signals it obtains, and this varies for each task it tries to learn. Unfortunately, most real-world problems that can be mathematically modeled have very sparse extrinsic rewards. This only reduces the fraction of useful samples in all the experience data that is collected by the agent, for it to learn meaningful policies from. As a motivating example, let us consider the task of solving a rubik's cube as a state space search problem (Agostinelli et al., 2019). Rubik's cube is a 3d puzzle with a total of over $4.3 \times 10^{19}$ possible states and 18 actions from each state that connects adjacent states. Despite having over 43 quintillion states, there's just one state that is considered solved, and only rewarding to reach that state truly defines the problem of solving the rubik's cube. It was only through rigorous search (with some

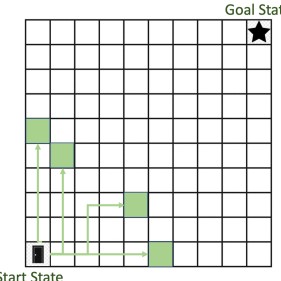 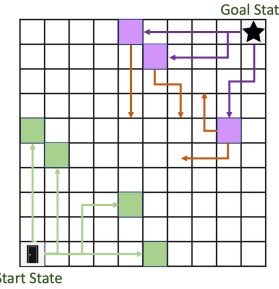 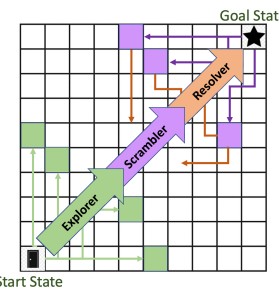

Figure 1: A visual depiction of (a) Hindsight Experience Replay trajectories (b) Bi-directional sample collection & (c) 'Scrambler-Resolver-Explorer' sample collection strategy with internal motivations. (Hindsight and Foresight goals are shown using light green and purple cells respectively)

state space alteration that was hand-crafted for this specific problem) that we were able to find that the shortest path is atmost 20 moves from any state to the goal/solved state.

One way to tackle this problem of sample efficiency is by careful introduction of implicit rewards and slight modifications that does not alter the original goal motive of the MDP formulation of the problem that is being solved. HER (Hindsight Experience Replay) (Andrychowicz et al., 2017) was a breakthrough in this context where it was able to slight modify the policy function used to incorporate, not just the useful samples but also the bad ones, for the agent to learn from. Specifically for the use cases where the reward is sparse and binary (similar to the rubik's cube case and most other state space search problems), this algorithm learns how to achieve alternative goals, which are essentially terminal states of failure trajectories. For their experiments, it was inferred that the agent performs better (faster) with learning multiple simultaneous tasks rather than just one. In our work, we extend this fact by using the agent to produce more simultaneous tasks by exploring the environment and exploiting the ability to sample goal states of the original task, and learn from them in parallel. Our main contributions in this paper include:

- A new Bi-directional Goal-conditioning (Binding Hindsight and the backward counter-part 'Foresight') formulation that defines a new reverse task $(\tau^{inv})$ obtained from the original state space search task $(\tau^*)$.
- A novel multi-task learning and sample generation strategy called 'Scrambler-Resolver-Explorer' (SRE) and its incorporation with the famous DQN algorithm.

## 1.1 AN INTUITIVE ILLUSTRATION

Figure 1 shows an example of our exploration method for collecting samples and goals in a 10 x 10 Gridworld for easy visualisation. The first figure (from the left) shows some trajectories obtained using the classical 'Hindsight Experience Replay' (HER) method with the task (we call $\tau^*$) of reaching the goal state of the environment. Here, the final goal of every unsuccessful trajectory is used to relabel the whole trajectory as the desired state (Hindsight Relabelling) and the agent is trained in a goal-conditioned manner. We also add these states to our candidates list (explained later). Despite learning from samples well, which otherwise wouldn't have been as fruitful, the algorithm still lacks an element that will help the agent push toward the goal more explicitly.

As part of our bi-directional goal-conditioning method, we add another spawn of the agent from the original goal state and run it for the backward task $\tau^{inv}$ (Here, we inform the desired final state for the agent is the original state state) to backward trajectories. Similar to the forward trajectories from HER, we take the end state of failed trajectories and add them to the candidate list as well as use them for Foresight Relabelling (Similar to Hindsight) and train the agent for this task. Here, we also define intermediate tasks similar to Ren et al. (2019). We take these intermediate candidates and sample them based on the ability of the agent to reach them, and reach the goal state of the original task from them (Intermediate Task Modification) and set them as goal states for a temporary task.

The intention to use an agent near the goal states is to use collect samples closer to the goal state, which might help in propagation of rewards through other regions better. But running the scrambler

collects samples for $\tau^{inv}$ and not $\tau^*$. To fix this, we start a third set of spawns from the end states of scrambler trajectories solving the original task (Trying to end in the goal state from $\tau^*$). Now with all these samples generated for different tasks ($\tau^*, \tau^{inv}$ and intermediate tasks), we adopt the Goal-Conditioned Setting and train a Universal Value Functional Approximator (Schaul et al., 2015) to be able to handle all these tasks. We believe this whole formulation of generating newer tasks, despite creating the overall learning objective of the value function more complicated, will generate more samples with rewards and thereby make the overall learning process efficient.

The visual depictions 1 above help draw a quick parallel with the classic graph traversal algorithm $A^*$. $A^*$ is a heuristic search algorithm that explores a graph by intelligently selecting the most promising paths based on an estimated cost-to-goal. While A* search relies on explicit knowledge about the problem domain, RL is data-driven and can handle more complex and dynamic environments just from the data collected by the agent. A traditional RL agent trying to collect experiences from the start state can be thought of as a vague random exploration on the graph since the reward is extremely sparse and it wouldn't have reached any sense of direction yet. We believe that these modules (Explorer, Scrambler and Resolver) that produce intermediate task re-definitions and using them for overloading the policy function could build an abstract internal directional heuristic that guides the agent not necessarily to a single goal, but also aids in learning paths with an overall sense of direction. (As shown in figure 1(c)) similar to the heuristic formulation in $A^*$. This way, we could possibly expect our Agent to develop internal abstract heuristic functions on any dynamic environment without having the need to manually handcraft potential functions to perform search.

## 2 RELATED WORK

**Bi-Directional Reinforcement Learning:** Our exploration algorithm is constructed around the concept of generating samples in a bi-directional manner (Start and Goal states). There has been very few papers that look into this style of bi-directionality. Edwards et al. (2018) generates training samples similar to ours but they try to learn the backward dynamics with a function approximator which will mathematically be dependent on the policy and might not be a good represent of the reverse MDP. Moore & Atkeson (1993) proposed to use model-based approaches for learning the environment dynamics first and then generating the predecessors that would help the agent converge faster. Recall Traces Goyal et al. (2019) pushed this ahead by proposing a generative model to perform backward trajectories from goal states. Florensa et al. (2017) is the next closest work, where the agent is spawned from states farther and farther away from the goal to ensure gradual learning. Most other bi-directional work [(Lai et al., 2020)] focuses on bi-directionality within each trajectory obtained to update their policies.

**Goal-Conditioning (GCRL) & Goal-Generation:** Kim et al. (2023) uses a GCRL setting Liu et al. (2022) and generates bidirectional curriculum similar to our work, but for a non-episodic setting and the reverse is usually within the trajectories. Ren et al. (2019) uses hindsight trajectories to generate new sub-goals to be used for intermediate task planning. Sharma et al. (2021) addresses the persistent goal-conditioned problem with a similar forward and reverse passes but also generates intermediate curriculum. Soroush Nasiriany (2019) uses a VAE for sub-goal planning in the latent space. Zhang et al. (2021) uses a classifier to predict the future state density following a certain policy and does planning using this to define intermediate/way-point states. We adopt a very similar approach to simplify the computation process by utilizing the policy network to generate intermediate goals/tasks.

## 3 PRELIMINARIES

**Reinforcement Learning:** In the context of Reinforcement Learning Algorithms, the problem is formalized as follows: an Agent, responsible for decision making, interacts with its Environment. This interaction is captured by a Markov Decision Process (MDP), where discrete time steps ($t = 0, 1, 2, ...$) govern the agent's and environment's dynamics. At each time step $t$, the agent receives a representation of the environment's state ($S_t \in \mathcal{S}$). Guided by a policy ($\Pi_t$), the agent selects an action ($A_t \in A(s)$). The agent obtains a reward ($R_t$) at each step, and the cumulative reward (Return) from a state is denoted by $G$. The objective is to find the optimal policy $\pi^*$ that maximizes the return ($G$) for the agent in each state. The agent's goal can also be rephrased as maximising the

value function $V^\Pi(s)$, which corresponds to the total expected discounted (by a factor of $\gamma$) reward as

$$V^\Pi(s) = \mathbb{E}_{s_0=s,a_t\sim\Pi(.|s_t),s_{t+1}\sim P(.|s_t,a_t)}[\Sigma_{t=0}^\infty \gamma^t R(s_t,a_t)]$$

**Q-Learning:** Q-Learning (Watkins & Dayan, 1992) is an off-policy reinforcement learning algorithm that determines the optimal action for each state. Through interactions with the environment, the agent updates Q-Values. Utilizing transition data, including the current state ($S_t$), current action ($A_t$), next state ($S_{t+1}$), and reward ($R_t$), the Q-Value corresponding to state $S_t$ and action $A_t$ is iteratively updated using the stochastic iterative update rule, which solves the Bellman equation:

$$Q^{new}(S_t, A_t) = Q(S_t, A_t) + \alpha(R_t + \gamma . \max_a Q(S_{t+1}, a) - Q(S_t, A_t))$$

Here, $\alpha$ denotes the learning rate. The policy, given Q-Values can be easily represented as $\Pi(s) = \arg\max_a Q(s,a)$

**Deep Q-Network:** For cases with large state spaces or high dimensionality of $\mathcal{S}$ function approximators can be used for representing the Q-function. In DQN Mnih et al. (2015) uses a deep neural network is used as function approximator. The input to the neural network is the current state, and the output is the corresponding Q-Values for each function. In DQNs, the update rule is $\theta_{t+1} \leftarrow \theta_t + \alpha[(R_t + \max_A Q(S_{t+1}, A; \theta_t) - Q(S_t, A_t; \theta_t))\nabla_{\theta_t} Q(S_t, A_t; \theta_t)]$ where the parameters of the neural network are denoted by $\theta$. To help with the convergence of the Q-Network used here, the algorithm suggests using two networks (Main Network that is updated every iteration and Target Network that gets updated with weights of the Main Network at a lower frequency). We use this stability factor in the DQN to define our custom learning agent.

**Goal-Conditioned RL:** In our work, we're mostly concerned about state space search problems. In such problems, this paradigm expects the agent to learn to navigate from any state to another, rather than sticking to a single or a small set of target states. In this setting, the reward function $\mathcal{R}$ is a binary indicator function to determine if the desired state (goal) is reached:

$$\mathcal{R}_{task}(s_t, a_t, s_t + 1) = \mathcal{R}_{task}(s_{t+1}, s_g) = \begin{cases} 1 & \text{if } s_{t+1} == s_g \\ 0 & \text{otherwise} \end{cases}$$

**Universal Value Function Approximators (UVFA):** Following the approach from (Schaul et al., 2015), where policies and value functions are trained on a goal $g \in \mathcal{G}$ and a state $s \in \mathcal{S}$, instead of the typical setting where it is just trained with states, Hindsight Experience Replay (HER) (Andrychowicz et al., 2017) proposes the policy network to be a function of the state and the goal (ie. $\Pi : \mathcal{S} \times \mathcal{G} \rightarrow \mathcal{A}$.) This way, the agent is expected to learn how to navigate from the start state to the goal state, both present in the inputs and the outputs can be used to find the best action. This reformulates the Q-function to depend on the goal in addition to the state-action pair $Q^\Pi(s_t, a_t, g) = \mathbb{E}[R_t|s_t, a_t, g]$. This also allows us to represent a single function approximator to represent a large number of value functions. Let a task $\tau : \mathcal{S} \times \mathcal{G} \rightarrow [0, 1]$ be the joint distribution over task starting state $s_0 \in \mathcal{S}$ and task goal $g \in \mathcal{G}$ and different goals, we could still use the same function approximator $V^\Pi(s_t, g)$ for various tasks by defining

$$V^\Pi(\tau^*) := \mathbb{E}_{(s_0,g)\sim\tau^*}[V^\Pi(s_o, g)]$$

As part of our work, we wish to extend this by generating more pseudo-goals that are close to the goal states of the original task. It would also help to have more samples in the vicinity of the goal state as that will train the policy network to converge to true Q-values faster. To achieve both these tasks, we define another agent that collects samples from the goal-state in reverse.

## 4    SCRAMBLER-RESOLVER-EXPLORER BASED DQN AGENT (SRE-DQN)

**Assumptions:** In this paper, we propose a new exploration strategy called "Scrambler-Resolver-Explorer" (SRE). We first make three practical assumptions that are applicable to most state space search problems. First, we assume the existence of goal states and our access to them. In many reinforcement learning applications related to control or path planning, goal states are defined and information about them is available. In cases where goal states are not explicitly defined, the agent can initially use a typical exploration strategy and then switch to the "SRE" exploration once it has discovered a subset of goal states.

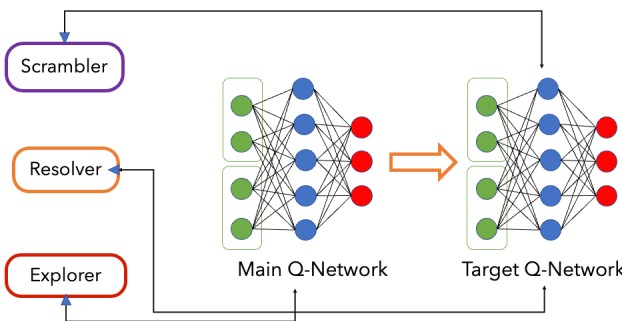

Figure 2: DQN: Scrambler-Resolver-Explorer Agent Architecture. Scrambler Policy & Resolver Policy are based on Target Q-Network, and Explorer Policy is based on Main Q-Network

Second, we assume that we can start a new environment from any state we provide, similar to the approach in (Florensa et al., 2017). The only difference is that we do not require any information on states closer to the goal state or a heuristic function in the state space. We simply expect to be able to spawn in desired states, such as the goal state. We believe this assumption is reasonable for most simulated environments.

Finally, we start with assuming that there exists a path from start state to goal state, as well as, from goal state to start state. As we increase the number of tasks (being able to start from state s' and reach state s"), we would require there to exist paths from any state to any other state. Since we don't require the shortest paths from state s to s' and backward to be the same, we do not make any assumptions about the reversibility of the MDP. Our fundamental assumption is that the state space is strongly connected. This is why we have restricted our study to strictly state space search problems (There exists a path from any state to another and the goal state set is a subset of the original state space).

**Agent Description:** In this particular study, since most well known state space search problems are puzzles with discrete action space, the algorithm chosen for implementation is DQN, although it is possible to combine this exploration algorithm with other off-policy algorithms such as NAC (Gu et al., 2016), DDPG (Lillicrap et al., 2019), SAC (Haarnoja et al., 2018a). We choose DQN because the features added to enhance the stability of convergence, such as experience replay and two policy networks with different update frequencies, are well-suited for the various agent definitions that we require. With this combination, we propose a novel DQN-like agent called SRE-DQN. Our algorithm has three sub agents that we call 'Scrambler' (S), 'Resolver'(R) and 'Explorer'(E). These modules try to collect samples in a bi-directional manner (Explorer generates forward samples and Scrambler generates backward trajectories from goal state) and our 'hindsight' setup helps convert this exploration method into a bi-directional learning strategy by potentially developing intrinsic motivation to bind transitions from both ends.

### 4.1 SAMPLE GENERATION

**Explorer:** This module represents the traditional RL agent component of this algorithm solving task $\tau^*$. We sample $(s_0, g^*) \sim \tau^*$ (with intermediate task redistribution explained later) agent starts from random a state sampled from the set of start states and tries out actions with an $\epsilon$-greedy fashion, with the hindsight feature on. This way, it can be explained that the essence of this module is to reach the final goal of the environment. When it acts greedily,it uses the Main Q-Network ($\mathbb{A}$) of the SRE-DQN agent. This policy is called $\Pi_{ex}$ and can be formally defined as:

$$\Pi_{ex:\{(s_0,g)\sim\tau^*\}}(s_i||g) = \begin{cases} \arg\max_a \mathbb{A}(s_i||g), & \text{with probability } (\epsilon) \\ \text{Random Action}, & \text{with probability } (1\text{-}\epsilon) \end{cases}$$

**Scrambler:** In an ideal scenario, for training the agent for task $\tau^*$, the knowledge of the MDP and its reverse dynamics could be used to ensure that the scrambler reaches states least explored thereby states that are most likely for the agent to have a bad value estimates. Florensa et al. (2018) tries

---

**Algorithm 1** SRE-DQN

---

**Given:**
- DQN-Agent with main Q-network $\mathbb{A}$ and target network $\mathbb{A}'$ with the Universal Value Function Approximation setting and Forward task $\tau^*$ of the environment.

Initialize $\mathbb{C}$ for candidates (Hindsight and Foresight) goals.
Initialize $\mathbb{A}$ & $\mathbb{A}'$ with same weights & replay buffer $\mathbb{B}$
**for** $Episode = 1$ **to** $M$ **do**
  **Explorer Module (I):** Sample $(s_0^*, g^*)$ from $\tau^*$
  Spawn agent at $s_0^*$ and run till end of episode with policy $\Pi_{ex}(.||g^*)$ to generate samples
  Add Hindsight Relabelled (With final state of episode) Samples to $\mathbb{B}$
  If not done: add final state to Candidate List $\mathbb{C}$ with ITR($\tau^*$) and ITR($\tau^{inv}$)
  **Explorer Module (II):** Repeat Explorer Module (I) with ($g \sim \mathcal{C}$) with ITR($\tau^*$)
  **Scrambler Module (I):** Sample $(s_0^{inv}, g^{inv})$ from $\tau^{inv}$
  Spawn agent at $s_0^{inv}$ and run till end of episode with policy $\Pi_{sc}(.||g^{inv})$ to generate samples
  Add Foresight Relabelled (With final state of episode) Samples to $\mathbb{B}$
  If not done: add final state to Candidate List $\mathbb{C}$ with ITR($\tau^*$) and ITR($\tau^{inv}$)
  **Scrambler Module (II):** Repeat Scrambler Module (I) with ($g \sim \mathcal{C}$) with ITR($\tau^{inv}$)
  **Resolver Module:** Spawn agent at $s_t$ (Final state of scrambler trajectory and run till end of episode with policy $\Pi_{re}(.||g^*)$ to generate samples
**end for**
**for** $t = 1, N$ **do**
  Sample minibatch $\mathbb{M}_B$ from replay buffer $\mathbb{B}$
  Perform one step of optimization on $\mathbb{A}$ using $\mathbb{A}, \mathbb{A}'$ and minibatch $\mathbb{M}_B$
  For every 'K' steps, update $\mathbb{A}'$ parameters with $\mathbb{A}$
**end for**

---

to achieve this by starting the agent from states closer to the goal state and eventually moving them farther and farther away and measuring its ability to solve from a state from the value function. But since we do not always have access to this from the simulator, in this paper, we try to condition the reverse problem of the original MDP as a new task $\tau^{inv}$. We rather use the same simulator to obtain empirical samples for $\tau^{inv}$ from $\tau^*$ by the following:

$$(s_0^*, g^*) \sim \tau^*, \quad \text{and} \quad (g^*, s_0) \equiv (s_0^{inv}, g^{inv}), \quad \text{where} \quad (s_0^{inv}, g^{inv}) \sim \tau^{inv}$$

Now that we have a new task for the same policy network, at the end of every episode, we initiate a new agent from the goal state $g^*$from$(s_0^*, g^*) \sim \tau^*$ equivalently, start state $s_0'$from$(s_0', g') \sim \tau^{inv}$ (with intermediate task redistribution) and run the agent to collect samples for this task. We use the Target network ($\mathbb{A}'$) to generate the samples for the scrambler module as generating scrambles is a very dynamic progress and making it dependent on the main Q-Network ($\mathbb{A}$) might make the whole learning process more unstable and affect the convergence of the policy network.

$$\Pi_{sc:\{(s_0,g)\sim\tau^{inv}\}}(s_i||g) = \begin{cases} \arg\max_a \mathbb{A}'(s_i||g), & \text{with probability } (\epsilon) \\ \text{Random Action} & \text{with probability } (1-\epsilon) \end{cases}$$

**Resolver:** The aim of adding a scrambler module is to ensure we collect enough samples in the region closer to the goal state for the training phase, but the above formulation generates samples for the backward task ($\tau^*$). Although the states and goals can be reversed to obtain samples that are inverted from the inverted task to simulate the forward task, the actions aren't directly reversible. To explain this better, let us say that the sample from $\tau^{inv}$ is $(s^1, a, s^2)$, to make this useful for the agent learning to solve task $\tau^*$, we would want $(s^2, a', s^1)$. To enable this task and the module itself to work, we require a function $revAction()$, which is hard to estimate without domain knowledge about the MDP. To overcome this requirement, we start another set of agents from every scrambler trajectory ($sc_{traj}$) end states and run them for the original task $\tau^*$ (with intermediate task redistribution) from here.

$$\Pi_{re:\{(s_0,g)|(s_0)\sim sc_{traj} and(\_,g)\sim\tau^*\}}(s_i||g) = \begin{cases} \arg\max_a \mathbb{A}'(s_i||g), & \text{with probability } (\epsilon) \\ \text{Random Action} & \text{with probability } (1-\epsilon) \end{cases}$$

**Intermediate Task Modification:** Revisiting the construction of intermediate task distribution from Ren et al. (2019), the first step is to obtain K trajectories from experience replay $\mathbb{B}$ that are most

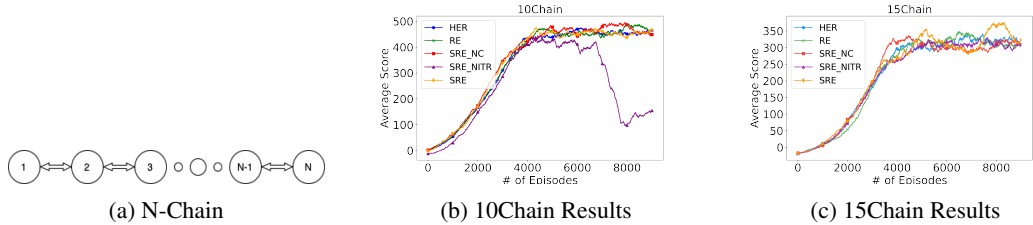

(a) N-Chain            (b) 10Chain Results            (c) 15Chain Results

Figure 3: NChain experiments: Performances of all variants of agents on environments with increasing state space size.

similar to that of the original task $\tau^*$. The expression below is the original objective re-written for the case where states and goals are in the same space (for state space search problems):

$$\Sigma_{i=1}^K \left( c||\hat{s}_0^i - s_0^i|| + \min_t \left( ||\hat{g}^i - s_t^i||_2 - 1/L \cdot V^\Pi(s_0^i, s_t^i) \right) \right)$$

where, $(\hat{s_0}^i, \hat{g}^i) \sim \tau^*$ and $\{(s_t^i)\}_{t=1}^T \sim \mathbb{B}$. Then, the surrogate goals for the intermediate task is chosen from $\arg\min_{s_t^i} \left( ||\hat{g}^i - s_t^i||_2 - 1/L \cdot V^\Pi(s_0^i, s_t^i) \right)$. In abstract terms, we could see that this procedure helps pick, first the trajectory that is closest to the original task $\tau^*$ and then the next steps finds the state in these trajectories that is closest to represent the goal from $\tau^*$. We adapt a similar idea but since the whole trajectory of the scrambler that is generated for the task $\tau^{inv}$ cannot be a good representative of the task, yet we want to utilize the scrambler trajectories. To make this work, we adopt the practical implementation idea from Zhang et al. (2021). In this work (Eysenbach et al., 2021), a classifier is learnt for predicting whether a state comes from a future state density following a policy or a marginal state, and then use this to classify way-points to use as surrogate goals. We stick to the regular distance metric similar to Ren et al. (2019). we consider all the terminal states of explorer and scrambler trajectories (hindsight and foresight goals) as candidates, we compute important weights for these candidates and sample them as surrogate goals for the corresponding task. Formalising this, let the task be $\tau$ (Could be $\tau^*$ or $\tau^{inv}$) and the set of candidates be $C$. Let the sample from the task distribution be

$$(s_0, g) \sim \tau \quad \text{Then,} \quad \text{importance}(c \sim C) \propto (d(s_0, c) \cdot \max_a Q(c, a, g)) / (d(c, g) \cdot \max_a Q(s_0, a, c))$$

Where $d(.,.)$ is some distance metric and $\max_a Q(s_1, a, s_2)$ represents the ability of the agent to go from state $s_1$ to state $s_2$ in our reward setting.

## 5 EXPERIMENTS

### 5.1 BASELINES

One of the main strengths of our method is how it can be viewed as an exploration strategy for sample collection that can be added to most off-policy algorithms. During training, Experience Replay-based methods for sample efficiency, such as Prioritized Experience Replay Schaul et al. (2016), Topological Experience Replay Hong et al. (2022), or any other methods, can be employed on the samples collected by the agents to train efficiently.

In this section, we focus on comparing the **SRE-DQN** (or DQN-SRE interchangably) with few relevant and similar methods derived to our case to study the impact of multi-directional goal conditioning and its effects on increasing state space size and convergence of the Q-function.

**HER-DQN:** uses DQN with hindsight experience replay relabelling. This will be a true baseline since this agent is only goal-conditioned in one direction (forward) and only collects samples from the start state (agent is not given information about the goal state).

**RE-DQN:** which uses goal conditioning in the forward direction but similar to Florensa et al. (2018) uses a random action agent to scramble from goal state and a resolver module starts from these scrambled states to obtain task $\tau^*$ samples making sure the samples obtained are of a similar spread (near start and goal state). This helps us understand the impact of uni-directional (forward) goal conditioning compared to multi-directional goal conditioning.

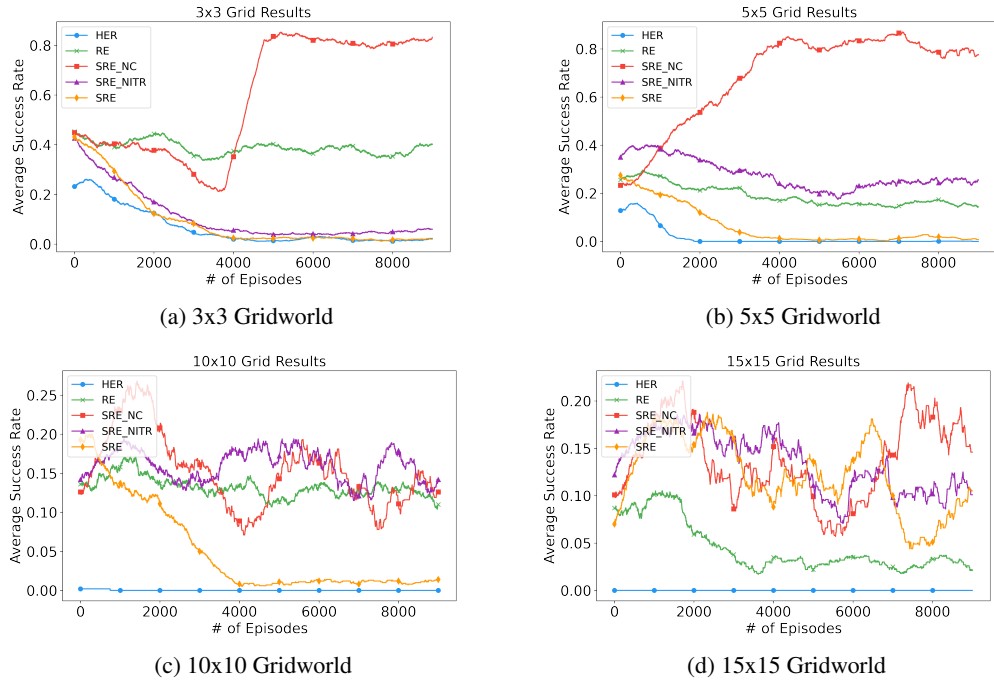

Figure 4: Gridworld experiments: Fewer-task agents (such as HER (1 task), RE (1 task), SRE_NC (2 task) perform better in smaller environments, but as the complexity of the environment increases, agents with more tasks (such as SRE_NC and SRE) perform relatively better.

**SRE_NC-DQN:** uses bi-directional conditioning similar to SRE-DQN but does not use any candidates in between. This essentially forces the agent to solely learn only two tasks (namely forward $\tau^*$ and backward $\tau^{inv}$) and reduce the burden of overloading multiple policy functions in one UVFA function.

**SRE-NITR**: is an SRE agent without 'Iterative Task Redistribution'. Here, we ignore the sampling method proposed (similar to way-point sampling from Zhang et al. (2021)) to sample from candidate goals in the state space. We still pick candidates in uniform, ensuring there are multiple modified tasks along with $\tau^*$ and $\tau^{inv}$ for the agent to learn from.

## 5.2 ENVIRONMENTS

We conducted our experiments on two classic state space search environments by increasing the size and goal-conditioning tasks. namely **NChain** 3a (10Chain and 15Chain) and **Simple Gridworld** (3x3, 5x5, 10x10 and 15x15). In the NChain problem , an agent moves along a chain of N states, with the goal of maximizing its cumulative reward. The agent can take two actions at each state: move forward to the next state, or jump back to an earlier state and each move has a probabilistic failure value = 0.2. The only positive reward in NChain emits at the rightmost terminal state (i.e., node N). We also used a simple Gridworld environment where the agent starts from the left bottom end and the goal state is top right. The agent is allowed to take 4 actions, one in any direction with probability of slip = 0.1 in any other direction.

## 5.3 SETTING

Since these search space problems are indeed simple environments for Deep RL agents, we choose to make the exploration and sample collection very strict (by reducing the number of episodes and number of steps in an episode lesser than typical RL experiments). To determine this experiment setting, we start with experimenting on the NChain environment (which is a flattened out version

of the Gridworld) since it is simpler to learn, and borrowed the same experiment setting to the Gridworld runs.

## 5.4 Observations

**NChain:** All agents learn to solve both the NChain environments well. However, the SRE_NITR agent, despite learning well for larger environments, did get unstable in the smaller environment. This suggests that introducing multiple intermediate tasks may lead to an overall increase in the number of meaningful samples, but may also pose the danger of sudden instability or may deter the convergence of the Q-Function.

**Gridworld:** With our experiment parameters, some agents' performances deteriorate with more episodes. This can be explained in light of the exploration factor ($\epsilon$-greedy) decaying with the number of episodes. Initially, these agents were able to reach the goal state due to random actions but were not reaching the goal state later since they did not learn the optimal policy. HER performs worse than the other agents in all variants of the Gridworld. On the other hand, SRE_NC, which is a purely bi-directional agent with two tasks ($\tau^*$ and $\tau^{inv}$) and no other intermediate tasks, performs well here. This suggests that *learning multiple tasks/goals with the same policy network can potentially help solve the forward task as well*.

Moreover, we observe a rather interesting trend in the relative performances of the different agents within a given Gridworld environment as a function of environment complexity (size) and agent 'complexity'. We can define agent 'complexity' to be proportional to the number of (sub)tasks (for instance, $\tau^*$, $\tau^{inv}$, and so on), that the agent learns in order to solve the environment. By this definition, HER and RE agents are low 'complexity' agents as they learn only one task, whereas SRE_NC, which learns two different tasks ($\tau^*$ and $\tau^{inv}$), is an agent with intermediate 'complexity,' and SRE_NITR and SRE, both of which learn more than two tasks (intermediate tasks), are high-'complexity' agents. The correlation between environment and agent complexity and the performance of the agents is discussed in further detail below.

- **Low 'Complexity' Agents:** As the complexity of the state space increases, we find that the performance of the low and intermediate-'complexity' (HER, RE, and SRE_NC) agents deteriorates.4.

- **High 'Complexity' Agents:** Interestingly, the trend mentioned above is not observed in high 'complexity' (SRE_NITR and SRE) agents. The performance of these agents is observed to increase as a function of the size of the state space, relative to the other lower 'complexity' agents.

The poor performance of the low and intermediate 'complexity' agents in more complex environments could be due to their capacity (which is a function of the agent 'complexity') for solving the environment having been reached. Meanwhile, high 'complexity' agents naturally need more samples to learn from. In our experiments, we had fixed the experiment parameters and only varied the state space size, which explains why less 'complex' agents (fewer tasks) perform better in less 'complex' state spaces with the same number of episodes and experiment parameters, and more 'complex' agents' relative performance gets better as we increase the state space complexity.

## 6 Conclusion

We introduce a novel Bi-Directional Reinforcement Learning (BDRL) algorithm called 'Scrambler-Resolver-Explorer' (SRE) that generates trajectories for forward task $\tau$, backward task $\tau^*$, and multiple intermediate tasks to generate more useful samples. Since BDRL (goal-to-state) is under-explored, we implement alternate versions of our model to compare and evaluate the individual modules. We demonstrate the advantage of learning multiple tasks/goals with the same policy network in solving the forward task. Furthermore, we observe a correlation between agent performance with state space size and number of tasks the agent learns. We believe that this work offers promising directions for further research in BDRL as well as goal-conditioning, particularly in exploiting state space related information (such as goal state knowledge) that the agent observes during its interaction with the environment.

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
