# OpenReview forum: "Bi-Directional Goal-Conditioning on Single Policy Function for State Space Search"
_ICLR.cc/2024/Conference — Submitted to ICLR 2024_

### Official Review · Reviewer_hi7c · 2023-10-30

**Soundness:** 3 good
**Presentation:** 2 fair
**Contribution:** 2 fair
**Rating:** 3
**Confidence:** 5

**Summary:**

This method proposes a framework called `SRE-DQN` which is a Goal-Conditioned Reinforcement Learning (GCRL) based framework where
goal states are sampled to create both a forward and a reverse task from the primary state space search. They introduce hindsight re-labeling for the forward tasks and a concept called `Foresight` for reverse tasks. To learn reverse moves from the reverse paths, agents are initiated from these paths to gather forward moves. All these components are glued together in a framework named `Scrambler-Resolver-Explorer (SRE)`.
Experiments are ran on toy problems with discrete state and action : `NChain` and a `Simple Gridworld`.

**Strengths:**

- This paper introduces an interesting exploration technique termed as `Scrambler-ResolverExplorer (SRE)`. I think the idea of learning forward and backward representations (i.e. learning how to reverse an action) is interesting in general. And this idea has similarity to [1] where a forward and backward representation of the reward is being learned.
- It is also a nice concept to try to have a distinction between an explorer, scrambler and resolver module where the primary focus of exploration of each module is distinct.





[1] Learning One Representation to Optimize All Rewards - https://arxiv.org/pdf/2103.07945.pdf

**Weaknesses:**

- The experimental results presented in Figure 3 and 4 are not convincing at all. The tasks `NChain` and `Simple GridWorld` are too simple and despite that the results seem very unstable.
- All the results are single-seeded and there is no measure of variance among multiple runs. I recommend the authors to present multi-seeded results over at least `10` random seeds per run to ensure reproducibility.
- `Figure 3` : All methods including the `HER` and `RE` baselines seem to work on-par and `SRE_NITR` diverges on the 10Chain. And on 15Chain all methods perform the same.
- `Figure 4` : Mixed and unstable results. The proposed method `SRE_NC` seems to have divergence issues.
- There has been more work for exploration in RL, looking at expanding trees and search - for example [1] which can potentially be a baseline.
- The framework still operates under a simple discrete states and action setting building on `DQN`. The authors claim that nothing is preventing
the framework to being extended to a continuous setting. However given the performance on the simpler setting, I'm not convinced it can readily be extended.
- HER has been tried on many grid world navigation tasks. I recommend to the authors to redo a literature survey on this to pull more related work.
- The writing and flow of the related work can be improved significantly. For instance, there is a related work section on `Surrogate Objective for Goal-conditioned RL` going through equations for the Lipschitz continuity assumption and Wassersetein distance and its not even relevant to their framework. Overall this makes reading the paper more difficult because I don't need to see the equations for related work.

[1] Probabilistic Planning with Sequential Monte Carlo methods - ICLR 2019 - https://openreview.net/pdf?id=ByetGn0cYX

**Questions:**

- Why is the HER baseline failing in Figure 4 ? If the reward is sparse and binary, HER should be resampling goals, why does it fail completely?
- Did the authros try other exploration mechanisms apart from `ϵ-greedy` ? For instance `Curiosity-driven Exploration` , or `Max-Entropy-RL` methods?

---

> ### Author Response · Authors · 2023-11-22
>
> Dear Reviewer,
>
> Thank you for your thorough and insightful review of our manuscript, "Bi-Directional Goal-Conditioning on Single Policy Function for State Space Search". We appreciate the time you took to evaluate our work and provide constructive feedback. Below, we address each of your points to clarify and improve our manuscript.
>
> **1. Experimental Results' Robustness (Figures 3 and 4):**
> We acknowledge your concern regarding the robustness of our experimental results. HER and other methods used here are well-established algorithms that can solve this state space search problem (which is a simple MDP) very well. Due to this reason, we’ve not added a best effort implementation. When all variants perform well, it might become hard to explicitly understand the performance of agents when using samples from a bigger task space. We observe this in Fig 3. NChain experiments. To differentiate the performances better, we use these experiment parameters (function approximator size, number of updates to the UVFA, exploration parameters, number of episodes and so on..) borrowed from NChain experiments to varying complexities of GridWorld. Fortunately, we were able to see clear differences in performance of the models and have reported the results. The code will also be made available under anonymity and a link to the same will be added to the abstract of the updated paper. Thank you for your suggestion on changing the experiment from single-seeded to multi-seeded results, we will modify our results based on this and update our results.
>
> **2. More Exploration Settings & Planning Literature**
> Thank you for pointing out some interesting literature. The primary research focus of our work hasn’t been to come up with the most optimal exploration/planning setting for state space search problems, but instead, just to check if defining new tasks from an existing task (if the original/forward task is to go from a start state to a goal state) like backward or intermediate tasks (start from goal state and go to the original start state/ or an intermediate state to another intermediate state) and overload all these tasks on a single policy function (by extending GCRL) helps the agent perform better in the original task. We’ve designed our baselines in a manner to exactly understand how different task definitions & formulations and training affects the performance in the main task.
>
> Thank you for pointing us to multiple exploration ideas. As part of our future work, we plan on extending this way of generating new tasks and training the agent, by combining it with various existing exploration and planning ideas.
>
> **3. Writing and Structure of Related Work:**
> We agree that the clarity of the related work section can be improved. We have revised this section, focusing on directly relevant literature and removing unnecessary technical details, such as the equations for the Lipschitz continuity assumption and Wasserstein distance. This revision would definitely make the paper more accessible and focused on our framework.
>
> Thank you again for your valuable feedback, which has undoubtedly helped improve the quality and clarity of our work.
>
> Best regards,
> Authors

---

### Official Review · Reviewer_mR2i · 2023-11-01

**Soundness:** 2 fair
**Presentation:** 2 fair
**Contribution:** 2 fair
**Rating:** 5
**Confidence:** 3

**Summary:**

The authors proposes Scrambler-Resolver-Explorer (SRE), which extends hindsight experience replay (HER) with bi-directional goal conditioning. SRE consists of three modules, Explorer, Scrambler, and Resolver for both the usual exploration and backward trajectory sampling from (original) goal states. It aims at gathering more samples close to the goal state region for more efficient training. They evaluate the proposed approach and compare it with baseline methods in NChain and GridWorld.

**Strengths:**

- The GCRL problem is an important problem, especially in terms of having controllability connecting the agent and its state in the environment.
- Given the employed assumptions, the proposed method that uses both directions for more effective GCRL is somewhat novel and might be useful.
- The manuscript is easy to read and follow.
- The baselines for the experiments are formed to examine the importance of some of the proposed modules.

**Weaknesses:**

- I believe the biggest weakness is that the empirical evaluation is done in simple environments, NChain and GridWorld. I believe that GCRL, which is about overcoming difficulties in reaching different goal states, needs environments with complexities in their dynamics to some degree (e.g., locomotion environments from MuJoCo) as testbeds.
- I have some concerns about the main assumption of the ability to spawn agents at arbitrary states, especially in the GCRL setting, where *reaching* specific goal states is the objective. Taking advantage of the simulated environments, if spawning at arbitrary states can be done without any costs, some combination of local exploration and spawning might be effective for both exploration and gathering various samples for re-labeling and goal-conditioned training.

**Questions:**

Please take a look at the weakness section.

---

> ### Author Response · Authors · 2023-11-22
>
> Dear Reviewer,
>
> Thank you for your constructive feedback on our submission “Bi-Directional Goal-Conditioning on Single Policy Function for State Space Search
> “. We appreciate your acknowledgment of the novelty and readability of our manuscript, as well as the importance of the Goal-Conditioned Reinforcement Learning (GCRL) problem.
>
> Regarding your concerns:
>
> 1. **Applicability to State Space Search Problems**:
>
> We build our idea on a fundamental assumption that if there is a path existing from start state to goal state, there is also a path back from the goal state to the start state. We tried to exploit the strength of GCRL, to train an agent to go from any state to any state, which implicitly leads to an assumption of action reversibility for each state-action pair. Unfortunately, this doesn’t hold true for typical RL benchmark environments. Due to this reason, we stick to typical state space search problems (problems like NChain and GridWorld)
>
> We agree that these environments (problems like NChain and GridWorld) used in our empirical evaluation are relatively simple. However, these simpler environments satisfy our reversibility assumption and also help us demonstrate our hypothesis of multi-task (start state -> goal state, goal state -> start state, intermediate states -> goal state and so on) definition learning and compare it with linearly increasing problem complexity. We acknowledge the need to test our approach in more complex environments, such as MuJoCo's locomotion tasks, which could better highlight the challenges and advantages in combining multi-task reformulations with GCRL. In future work, we plan to extend our evaluation to these more complex scenarios (that still remain reversible) to further validate the effectiveness of our approach.
>
> 2. **Assumption of Spawning Agents at Arbitrary States**:
>
> It is true that the ability to spawning at arbitrary states when combined with the GCRL setting will definitely improve the quality of samples obtained (after relabelling) and this is what we wish to fundamentally explore. Although, to prevent any unplanned correlations between the task definitions, we don’t allow the agents to explicitly remember/control the spawning state sets.
>
> Our primary research objective in this work has been to check, if the multiple different tasks defined during the exploration phase (starting from any state s’ and reaching new state s’’) generating more samples with rewards, enables the agent to learn more about the whole environment better, and if this knowledge will help in solving the primary task (start state and goal state of the original state space search problem) and from our set of experiments, there seem to be some potential improvement which deserves further and in-depth study. We believe this ideation and formulation of the GCRL problem will be further broken-down and studied by the community.
>
> Thank you once again for your insightful feedback and for pointing out these critical aspects of our research.
>
> Sincerely,
> Authors

---

### Official Review · Reviewer_bwK3 · 2023-11-05

**Soundness:** 1 poor
**Presentation:** 2 fair
**Contribution:** 1 poor
**Rating:** 3
**Confidence:** 4

**Summary:**

The paper proposes a way to sample and relabel goals to increase the sample efficiency of an off policy goal reaching agent. To do this, the authors propose three different state-goal samplers. The explorer, the scrambler and the resolver.

The explorer tries to solve the original task of reaching desired goals from the starting states.  The scrambler inverts this problem. The scrambler is started from the desired goal states of the original task and tries to reach the starting states of the original task. The resolver samples subgoals or waypoints which balance two objectives:
1) are reachable from the original start states and,
2) the original goals are reachable from them.

The authors instantiate this method using a DQN agent and perform experiments on many grid world MDPs.

**Strengths:**

The idea of sampling waypoints can be very beneficial for sample efficient goal reaching. The paper recognizes this correctly.

**Weaknesses:**

Weakness:
1) It seems as if there is an implicit assumption (which should be explicitly stated) that the environment is reversible. That is, the start states can always be reachable from the goals states. This is not always true, for example you can't "uncook" food.
3) Theoretically, there is no proof provided that the sampling method proposed can learn a meaningful goal conditioned policy. I understand it is difficult to provide any theoretical guarantees for relabeling methods, and perhaps this is out of scope of the paper. But, see the next point.
4) All experiments are performed on extremely simple toy MDPs where judging the benefits of a complex goal sampling technique can be difficult. Moreover, the proposed method SRE-DQN doesn't perform the best in any of the tasks.
5) It is surprising that the success rate on such simple MDPs is lower than 20%. For example in a 3x3 MDP, both the SRE-DQN and HER get around 0 success rate. I suspect that an error in the implementation is causing this.
5) Implementation details as well as the code have not been provided.

**Questions:**

Suggestions for improvement and minor questions:
1) After reading just Section 1, it is unclear how the agent can collect trajectories starting from the goal state. If this is a assumption that the authors are making, then they should state it clearly in the introduction. They should also state why this assumption makes sense, and what are its limitations.
2) In Section 4, 2nd para the authors state : "Scrambler generated backward trajectories from goal state adversarial to the explorer". Why are the goal state adversarial to the explorer?
3) What are "state space search" problems? Is this the same as goal conditioned RL problems where the goal space is equal to the state space?


Typos:
1) Section 2, 2nd para: simiilar -> similar
2) Section 3, 4th para: In the reward definition, $+1$ should be in the subscript.
3) Section 4, 3rd para : generated -> generates
3) Section 5.1, 4th para:  multi-directional) -> there is no corresponding opening bracket.
4) In Section 3, para Surrogate Objective for Goal-conditioned RL:  $\tau$ -> $\tau^*$

---

> ### Author Response · Authors · 2023-11-22
>
> Dear Reviewer,
>
> Thank you for your constructive feedback on our submission “Bi-Directional Goal-Conditioning on Single Policy Function for State Space Search “. We appreciate your acknowledgment of the novelty and readability of our manuscript, as well as the importance of the Goal-Conditioned Reinforcement Learning (GCRL) problem.
>
> **Assumption of Environment Reversibility**
>
> Our main assumption would be that there exists a path between goal state and start state (essentially, every new task we define, there exists a path from the start state and goal state of that particular task). The point you’ve mentioned is very true, since as we increase the number of tasks, our assumption does directly converge to reversibility of state-action pairs. We acknowledge that we did not explicitly state this assumption. Thank you very much for pointing that the point is missing and the necessity to make it clear and explicit. We will add a small paragraph to discuss this limitation & assumption and make it explicit. We will also add a small part explaining how, these specific MDPs where path from any state to another and the goal state set is a subset of the original state space, are what we call ‘state space search problems’ in our work.
>
> **Environments, Performance of SRE-DQN not being the best & relative poor performance of other agents:**
>
> We agree that a broader range of experiments is necessary. We agree that these environments (problems like NChain and GridWorld) used in our empirical evaluation are relatively simple. However, these simpler environments satisfy our reversibility assumption and also help us demonstrate our hypothesis of multi-task (start state -> goal state, goal state -> start state, intermediate states -> goal state and so on) definition learning and compare it with linearly increasing problem complexity.
>
> We acknowledge the need to test our approach in more complex environments, such as MuJoCo's locomotion tasks, which could better highlight the challenges and advantages in combining multi-task reformulations with GCRL. In future work, we plan to extend our evaluation to these more complex scenarios (that still remain reversible) to further validate the effectiveness of our approach.
>
> **Regarding Low Success Rates:**
>
> As you have mentioned, HER and other methods used here are well-established algorithms that can solve this state space search problem (which is a simple MDP) very well. Due to this reason, we’ve not added a best effort implementation. When all variants perform well, it might become hard to explicitly understand the performance of agents when using samples from a bigger task space. We observe this in Fig 3. NChain experiments. To differentiate the performances better, we use these experiment parameters (function approximator size, number of updates to the UVFA, exploration parameters, number of episodes and so on..) borrowed from NChain experiments to varying complexities of GridWorld. Fortunately, we were able to see clear differences in performance of the models and have reported the results. The code will also be made available under anonymity and a link to the same will be added to the abstract of the updated paper.
>
> **Regarding SRE’s Performance:**
>
> Our primary research focus in this paper has been about generating new ‘task’ definitions within the same state space, training the model on these newly defined tasks, and seeing if that helps the agent in solving the primary state space better. The full SRE agent uses forward, backward and multiple intermediate task definitions (multi-directional). Following the main claim of the paper (bi-directionality/two-task definitions), we see that SRE_NC (variant that only uses bi-directionality and does not use intermediate candidates) seems to be working well for our experiment setting.
>
> **Other Suggestions & Corrections**
>
> Since the UVFA acting as a policy function takes in two inputs (current state (s1) and desired state (s2)) and provides the optimal/desired action, to generate backward trajectories, we start the agent from the original goal state and provide the original goal state as the desired state into the policy function (UVFA). This way, in an abstract manner, we can say that we are requesting the agent to start from the goal state and traverse towards the start state. I understand this might have not been clear in section 1, in our new iteration that will be uploaded soon, I will make amends to the Section 1.1 to explain this. Thank you so much for pointing this out.
> We apologize for the oversight. Producing adversarial trajectories would require exploiting the current policy (from forward) but we do not perform any such action.We have removed the part about the Scrambler being adversarial to the forward trajectories.
>
> We look forward to the opportunity to contribute to this field. Thank you for your valuable feedback.
>
> Sincerely,
> Authors

---

> ### Comment · Reviewer_bwK3 · 2023-11-22
> **Response to authors**
>
> Dear authors,
>
> Thank you for acknowledging my concerns.
>
> > Our main assumption would be that there exists a path between goal state and start state
>
> Is this the same as assuming reversibility of the MDP? Reversibility: Given that you can reach a goal state from a start state in an MDP, the assumption is that you can also reach the start state from the goal state.
>
> > However, these simpler environments satisfy our reversibility assumption and also help us demonstrate our hypothesis of multi-task
>
> You could also extend the same toy MDPs to be more complex by increasing the size of the state-space. The example of solving a rubic's cube is, which you have included in the introduction can be a good task to test the improvement of your algorithm.
>
> > we’ve not added a best effort implementation
>
> Instead of this, I think you should make the tasks more difficult. If you don't find a difference in 10x10 grids, then consider making the grid 100x100.

---

> > ### Author Response · Authors · 2023-11-23
> > **Regarding Reversibility Assumption**
> >
> > Dear Reviewer,
> >
> > Thank you for your suggestions.
> >
> > Yes you are right with "Given that you can reach a goal state from a start state in an MDP, the assumption is that you can also reach the start state from the goal state." But the catch here is that, we don't really require the paths (forward and backward) to be the same. This way, we don't require each and every MDP state-action pair to be reversible (In this context, I mean, if there exists an action a that takes the agent from state s to s', there is an action a^{inv} that takes the agent from s' to s.)
> >
> > We just need there to exist a path between any two states. So our fundamental assumption would be that our state space is strongly connected.

---

### Meta-Review · Area_Chair_sPTn · 2023-11-30

**Metareview:**

**Summary**: The paper proposes a goal-sampling method for goal-conditioned RL aimed at improving sample efficient. The method combines three strategies: reaching the goal from the initial state, reaching the initial state starting at the goal, and starting at some waypoint between the start and the goal. Experiments on two tasks demonstrate the efficacy of the method.

**Strengths**: Reviewers agreed with the importance of the goal-conditioned RL problem, and generally liked the intuition behind the proposed method.

**Weaknesses**: Reviewers were concerned that the proposed method makes assumptions not made by prior methods (e.g., that the dynamics are reversible, that the agent can choose its initial state). A second concern was about the thoroughness of the results: only 2 relatively simple tasks where used, and the results were sometimes inconclusive. Reviewers also suggested extending the paper to include more implementation details and theoretical justification.

**Justification For Why Not Higher Score:**

All reviewers voted to reject the paper (scores = 3, 3, 5) because of limited experiments and for making assumptions not made by prior methods.

**Justification For Why Not Lower Score:**

N/A

---

### Decision · Program_Chairs · 2024-01-16

Reject